# Encouraging Individuals to Adapt to Climate Change: Relations between Coping Strategies and Psychological Distance

**Mary Guillard \*, Ghozlane Fleury-Bahi and Oscar Navarro** 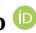

Laboratoire de Psychologie des Pays de la Loire (LPPL–EA 4638), Faculty of Psychology, University of Angers, University of Nantes, F-44000 Nantes, France; ghozlane.fleury@univ-nantes.fr (G.F.-B.); oscar.navarro@univ-nantes.fr (O.N.)
\* Correspondence: consultante.maryguillard@gmail.com

**Abstract:** Experts agree that the environmental situation in relation to climate change requires that populations mobilize. In this respect, research on psychological distance shows that the fact of perceiving an event as concrete leads individuals to adapt to this environmental issue. The first aim of this research study is to identify the different types of environmental coping as regards climate change. The second objective is to study the relations between psychological distance relative to climate change and environmental coping strategies via a quasi-experimental protocol. In order to do this, 345 participants were assigned to a group where climate change was presented as more or less distant from a spatial, temporal, social or hypothetical point of view. On the one hand, the results enable the identification of two second-order factors regarding coping strategies in relation to climate change: Strategies centered on accepting climate change and those centered on minimizing its gravity. On the other hand, covariance analyses and path analyses show that, in general, a small psychological distance in relation to climate change is likely to be associated with more strategies centered on accepting climate change and fewer strategies focused on minimizing its gravity. This study leads us to ponder the pertinence of considering the psychological distance model, notably during awareness-raising campaigns.

**Keywords:** climate change; construal-level theory; coping strategies; psychological distance

## 1. Introduction

Individuals perceive the threat of climate change (CC) as distant in time and space (Lorenzoni & Pidgeon, 2006; Pidgeon, 2012) [1,2]. Given its seriousness and anthropogenic cause, it seems essential for individuals to mobilize in order to adapt to this environmental phenomenon [3]. In this regard, 79% of French people say that climate change is caused by human activities and global warming is the most acute [4]. In that sense, 44% of French people believe that individual behavior is an effective solution to fight against global warming [5]. As climate change preoccupations are related to how individuals adapt to it [6], it is important to understand how the way in which the phenomenon is characterized influences adaptation strategies.

### 1.1. Evaluation of Climate Change: The Psychological Distance Model

The model of psychological distance is derived from construal-level theory [7,8]. According to this theory, the individual makes a mental representation of an object according to its degree of abstraction. This representation is thus situated at higher or lower construal levels, depending on the degree of abstraction of the considered object. Psychological distance depends on this degree of abstraction and is modulated by four interdependent psychological barriers: Spatial, social, temporal, and the uncertain nature of the object. Regarding environmental problems, individuals tend to assess negative effects as being distant from viewpoints that are temporal [9–11], spatial [9,10,12], and social [13,14]. In addition, many individuals do not seem to question the uncertain nature of CC [5,15]. In

other words, they believe in the reality of climate change. Simultaneously, 17% of French people declare that disasters are essentially natural and not necessary linked to anthropogenic climate change [4]. It seems that links between climate change, consequences and causes are not really clear.

These various elements underline the idea that environmental problems, especially those related to CC, are generally perceived as being distant, or even abstract [16,17]. This could be linked to the fact that individuals find it difficult to establish the link between the causes of CC, the phenomenon itself, and its repercussions [17,18]. Considering that the abstract or concrete nature of an object could influence decision making [8], it seems pertinent to study the relationships between psychological distance regarding CC and the way people adapt to it.

### 1.2. Reducing Psychological Distance in Order to Encourage Individuals to Adapt to Climate Change?

Psychological distance barriers are relevant to analyze how individuals adapt to CC [19]. However, relations between psychological distance and adaptation are unclear [20–22]. On the one hand, perceiving CC as concrete is associated with a higher level of concern and with more pro-environmental intentions [23–25]. For instance, taking into account the impacts of CC physically near is associated with more behaviors aiming to reduce greenhouse gas emissions [26] as well as with a greater environmental concern [27]. The social barrier, designating the distance between self and others, could also explain adaptive strategies to climate change [24,28]. In particular, research has shown that asking individuals to put themselves in the position of a victim of CC leads to more pro-environmental intentions [28]. Lastly, the uncertain nature of a situation could make the implementing of adaptive strategies more difficult [15,29,30]. Likewise, when CC is perceived as in progress, individuals are more likely to support environmental policies or to have pro-environmental behaviors [14,31]. On the other hand, some authors suggest that perceiving proximal CC do not systemically lead to pro-environmental intentions or behaviors. In that sense, individuals report more consistent pro-environmental intentions when they think about the long term [32,33]. Thus, a strong temporal distance could be associated to higher levels of constructs, referring to individual values for example [7,8,20,32]. In addition, the perceived threat induced by a proximal CC could also be associated to coping strategies not necessarily adapted to the CC [20].

This lack of consensus highlights the relevance to study in what extent psychological distance or proximity are associated with individual adaptation to climate change, or not. In that sense, it seems interesting to consider adaptation to face an environmental threat.

### 1.3. Adapting to Climate Change: Coping Strategies

Environmental threats can be associated with stressors [34–36]. Thus, to adapt and deal with a stressful situation, individuals can implement coping strategies [37–40]. There would seem to be two sets of coping strategies that can be used: One centered on the problem, which generally represents active coping (seeking information, finding support from professionals, etc.) and the other focusing on emotional regulation, which tends to illustrate passive coping (dramatizing the situation, circumventing the problem, etc.) [39,41,42]. While the first strategies seem to be used when the situation seems controllable, the latter are more likely to be implemented when the situation seems uncontrollable [41]. In addition, some authors identify specific strategies for certain situations [34,35], especially when it comes to environmental threats [35,42,43]. More specifically, two large sets of coping strategies can be used: Problem-centered strategies (expressing emotions, problem-solving), and those that are distanced from the problem's center (relativization, denial of guilt, pleasure) [35]. For more details about these coping strategies, see the questionnaire in supplementary presenting the associated items. In fact, coping refers to a dynamic adaptive process that depends on environmental changes and the way the individual perceives them [43]. In that sense, the assessment of a situation perceived as a threat leads to coping strategies [38,39,44–46]. Lastly, it seems that coping strategies play a mediation

role between the manner of assessing the environmental situation and the producing of pro-environmental behavior [35,47].

*1.4. Objectives and Hypotheses*

To our knowledge, there is no study about the relations between the way people evaluate CC, taking into account simultaneously the four barriers of psychological distance, and the implementing of coping strategies. This research aims to identify the coping strategies implemented regarding CC, and to understand how psychological distance in relation to CC, as well as its sub-dimensions, influences these coping strategies. In order to do so, a quasi-experimental methodology was preferred in the aim of identifying the influence of each of the dimensions of psychological distance on the coping strategies related to climate change. Two general hypotheses composed this study: (1) Two types of coping strategies related to climate change can be identified, and (2) the psychological distance barriers of climate change predict these coping strategies. The following hypotheses are put forward:

- Given that coping strategies refers to two large sets of strategies, made up of various types of coping [35,39,41,42], we hypothesize that coping regarding CC is also composed of two second-order factors that illustrate two sets of coping strategies (H1). One refers to coping composed of problem-centered strategies, while the other designates coping made up of strategies that are distanced from the center of the problem.
- The way in which CC is presented influences the perception of the global phenomenon [23,48], as well as the way of facing it [23,28]. Indeed, the manner of presenting CC influences psychological distance in relation to CC (H2a) and the associated coping strategies (H2b). Environmental intentions and concerns seem to be related to psychological distance [23,24], we hypothesize that psychological distance barriers and coping strategies are also related (H2c).

## 2. Methodology

*2.1. Participants and Procedure*

For this study, participants were recruited through forums and online ads (Facebook and Twitter), from September 2017 to November 2017. We recruited a total sample of 345 people. Only adult participants (over 18) living in France were included. The participants were invited to fill out an online questionnaire; 82.6% of the total sample were women and 17.4% were men, with the mean age standing at 26.70 years (*SD* = 10.20). Furthermore, 56.8% of the participants were students, 30.7% had a professional occupation, and 7.2% were in another situation.

Eight experimental groups and one control group were created. The individuals were randomly assigned to one of the groups. When the participants are assigned to one of the eight experimental conditions, they watched a video presenting climate change as more or less distant and focusing on one of the four aspects of psychological distance (spatial, temporal, social, or uncertain). Regarding the control group, climate change was presented without it being more or less distant, and without focusing on one of the four dimensions of psychological distance.

In order to verify our experimental manipulations, participants were invited to answer a question regarding the video they had watched. These experimental verifications led us to draw up a sub-sample for analyzing the influence of experimental conditions. These experimental verifications are detailed in Section 2.2.2. Among the 345 people recruited, 286 were considered for studying the influence of experimental conditions. Thus, 87 participants were assigned to the conditions where CC is presented as distant (between 21 and 24 participants per aspect emphasized), and 108 participants were assigned to the control group. When the distribution of the participants given eight experimental conditions and the control condition is examined, no significant difference regarding gender is observed ($\chi^2$ (8) = 6.23, *p* = 0.62). However, an unequal distribution between the nine groups regarding age is noted ($\chi^2$ (24) = 56.40, *p* = 0.01), with the under-26-year-olds



overrepresented in the control group. No unequal age distribution is noted when only the experimental groups are considered ($\chi^2$ (21) = 31.62, *p* = 0.08).

*2.2. Material*

For this experimentation, participants were invited to watch a video (Videos are available on www.youtube.com/playlist?list=PLvELfPmXB897-JyvoJEKmFq3L_r2ZMNWG) and to answer an online questionnaire and were assigned randomly to one of the conditions. The sociodemographic questions were asked at the end of the experiment.

2.2.1. Framing Videos

The videos that were presented lasted on average 219 s (*SD* = 2.83) and were composed of four parts: A definition of CC ($M_{duration}$ = 12 s, *SD* = 0.00), causes of CC ($M_{duration}$ = 20 s, *SD* = 0.00), temperature increase ($M_{duration}$ = 56 s, *SD* = 1.19), consequences of CC (floods, droughts) ($M_{duration}$ = 131 s, *SD* = 3.23).

When the focus is on the temporal aspect, the videos feature the expected consequences of CC in the next few years in the case of large psychological distance (example: Temperature forecast in 2050), or in the case of lesser psychological distance (example: Observed droughts in 2015 and 2016). The videos focusing on the social aspect show the repercussions of CC on individuals living in extreme precariousness in both the case of significant psychological distance (examples: Testimonials from people living in precariousness during a heatwave or droughts) or of reduced psychological distance (examples: Testimonials from people not living in precariousness during droughts). Videos focusing on the spatial aspect highlight the impact of CC outside Europe when the psychological distance is large (examples: Droughts in India or Australia) and the impact of it as observed in mainland France, in the case of smaller psychological distance (examples: Droughts in France and Germany). Lastly, when the focus is on the uncertain nature of the phenomenon, the videos show discourses establishing the link between CC and its effects, in the case of reduced psychological distance (example: Climate change leads to more heat weaves), or discourses that challenge this relation (example: Droughts explained by water resource management or intensive agriculture) in the case of greater psychological distance.

2.2.2. Manipulation Check

The aim of this measurement is to verify the validity of the experimental conditions. Thus, we made sure that the participants had indeed perceived that the video they were shown focused on one of the aspects of psychological distance (spatial, temporal, social, or uncertainty). This measurement thus corresponds to an item regarding the video. The individuals were asked to indicate the degree to which they agreed on a Likert scale, ranging from 1 ("Strongly disagree") to 5 ("Strongly agree"). When the participant's answer was congruent with the video, they were included for the analyses studying the video's effect. For instance, when we showed the video that presented CC as near, from a spatial point of view, we asked the participant assigned to this condition if the video he had watched showed the effects of CC observed in mainland France. The participant was only included if he agreed with the proposition (answers 3, 4 or 5), and the participant was removed if he was not agreed (answers 1 or 2).

2.2.3. Psychological Distance Scale

A psychological distance scale linked to CC was used. Originating from the work of Jones et al. (2017) [23], this scale is composed of 16 items (example: "When I think about the effects of climate change, I think about distant countries"). The participants answered the items by positioning themselves on a Likert scale ranging from 1 ("Strongly disagree") to 5 ("Strongly agree"). The scale was translated into French by two bilinguals (English-French). A factorial analysis was conducted on the 345-participant sample. This analysis led to remove four items (items 1, 3, 4, 5; see Supplementary Materials) and the obtaining of three dimensions: The social and temporal barriers ($\alpha$ = 0.78; items 8, 9, 10,

13, 14, 15), the spatial barrier ($\alpha$ = 0.67; items 6, 11, 16), and the uncertain nature of CC ($\alpha$ = 0.61; items 2, 7, 12). Contrary to Jones et al.'s analyses [23], we observe in this case that the dimensions relative to the social and temporal barriers merged. The reliability of these dimensions is acceptable because a Cronbach's alpha of 0.60 is enough when there are fewer than 10 items [49]. The reliability of the global psychological distance score, calculated on the basis of the twelve remaining items, is satisfactory ($\alpha$ = 0.75).

### 2.2.4. Scale of Coping Strategies

In order to measure the coping strategies, the scale proposed by Homburg et al. [35] was used. In total, 37 items were proposed to the participants, who needed to answer using Likert scales ranging from 1 ("Never") to 5 ("Always"). These items were adjusted to the theme of climate change and translated by two bilinguals (English-French). The factorial analysis conducted on the 345-participant sample led to the withdrawal of nine items (items 3, 14, 23, 24, 26, 29, 30, 31, 35; see Supplementary Materials), and enabled identification of the same dimensions as Homburg et al. [35]: Expression of emotions ($\alpha$ = 0.78; items 21, 33, 36), problem-solving ($\alpha$ = 0.90; items 6, 9, 12, 17, 22, 25, 27, 37), wishful thinking ($r$ = 0.42, $p < 0.01$; items 5, 11), denial of guilt ($\alpha$ = 0.84; items 4, 10, 13, 16, 28, 32), relativization ($\alpha$ = 0.78; items 8, 18, 34), resignation ($\alpha$ = 0.71; items 1, 2, 19, 20), and pleasure (2 items, $r$ = 0.57, $p < 0.01$; items 7, 15).

### 2.3. Data Analyses

The analyses were made on SPSS and AMOS, 23rd version. Firstly, a structural equation model analysis was conducted in order to identify the two sets of strategies (second-order factors) composed of the various types of coping strategies (first-order factors) (H1). In order to verify the model's adjustment, the following recommendations were respected [50–52]: $\chi^2$= ns or $\chi^2/df$ between 1 and 3; GFI > 0.90; CFI > 0.90; RMSEA < 0.08; SRMR < 0.08. Secondly, an analysis of the general linear models (e.g., ANCOVA) was performed to verify if the experimental conditions could explain the scores obtained on the psychological distance scales related to CC (H2a) and on the environmental coping scales (H2b). We took into account the Partial Eta-Squared ($\eta p^2$) in order to consider the effect size, knowing that 0.01, 0.06, and 0.14 correspond respectively to low, average and high effects [53]. Thirdly, a causal research model analysis was made in order to study the relations between the psychological distance barriers and the coping strategies, all the while taking into account the influence of the experimental conditions (H2c). While the first model includes the two sets of coping strategies, the second takes into account the seven types of coping strategies studied. In order to verify the adjustment of these models, the previous recommendations were followed.

## 3. Results

### 3.1. Analysis of the Factorial Structure of the Coping Scale

As a reminder, Homburg et al. [35] identified two second-order factors regarding problem-centered strategies (expression of emotions, problem-solving) and strategies distanced from the center of the problem (denial of guilt, relativization, pleasure). As their second-order model does not include the strategies of wishful thinking and resignation, we tried to integrate them to our own. Table 1 displays the correlations between the scores relative to the various coping strategies that are considered for identifying the second-order factors. In order to verify the model's adjustment, these recommendations were followed [50–52]: $\chi^2$= ns or $\chi^2/df$ between 1 and 3; GFI > 0.90; CFI > 0.90; RMSEA < 0.08; SRMR < 0.08. The model analysis does indeed enable identification of two second-order factors, corresponding to the same structure as Homburg et al.'s. (2007). While the model is correctly adjusted when wishful thinking is included as a problem-centered strategy ($\chi^2$ (231) = 475.61, $p < 0.01$; $\chi^2/df$ = 2.06; GFI = 0.90; CFI = 0.93; RMSEA = 0.05; SRMR = 0.07), it is not adjusted when we integrate the dimension of resignation as a strategy that is distanced from the center of the problem ($\chi^2$ (331) = 763.88, $p < 0.01$; $\chi^2/df$ = 2.31;

GFI = 0.87; CFI = 0.88; RMSEA = 0.06; SRMR = 0.09). Figure 1, which corresponds to the identified model, clearly pinpoints two second-order factors regarding coping in relation to CC (H1). Lastly, the reliability index is satisfactory for the general factors of problem-centered coping strategies ($\alpha$ = 0.85) and strategies that are distanced from the center of the problem ($\alpha$ = 0.81).

**Table 1.** Pearson's correlation coefficients between the scores relative to the various coping strategies.

|  | 1 | 2 | 3 | 4 | 5 | 6 | 7 |
|---|---|---|---|---|---|---|---|
| 1. Expression of emotions | _ |  |  |  |  |  |  |
| 2. Problem-solving | 0.52 ** | _ |  |  |  |  |  |
| 3. Wishful thinking | 0.27 ** | 0.14 ** | _ |  |  |  |  |
| 4. Denial of guilt | −0.26 ** | −0.29 ** | −0.05 | _ |  |  |  |
| 5. Relativization | −0.17 ** | −0.19 ** | 0.12 * | 0.23 ** | _ |  |  |
| 6. Pleasure | −0.44 ** | −0.36 ** | −0.08 | 0.29 ** | 0.23 ** | _ |  |
| 7. Resignation | −0.01 | −0.15 ** | 0.07 | 0.14 ** | −0.35 ** | −0.03 | _ |

** $p < 0.01$; * $p < 0.05$.

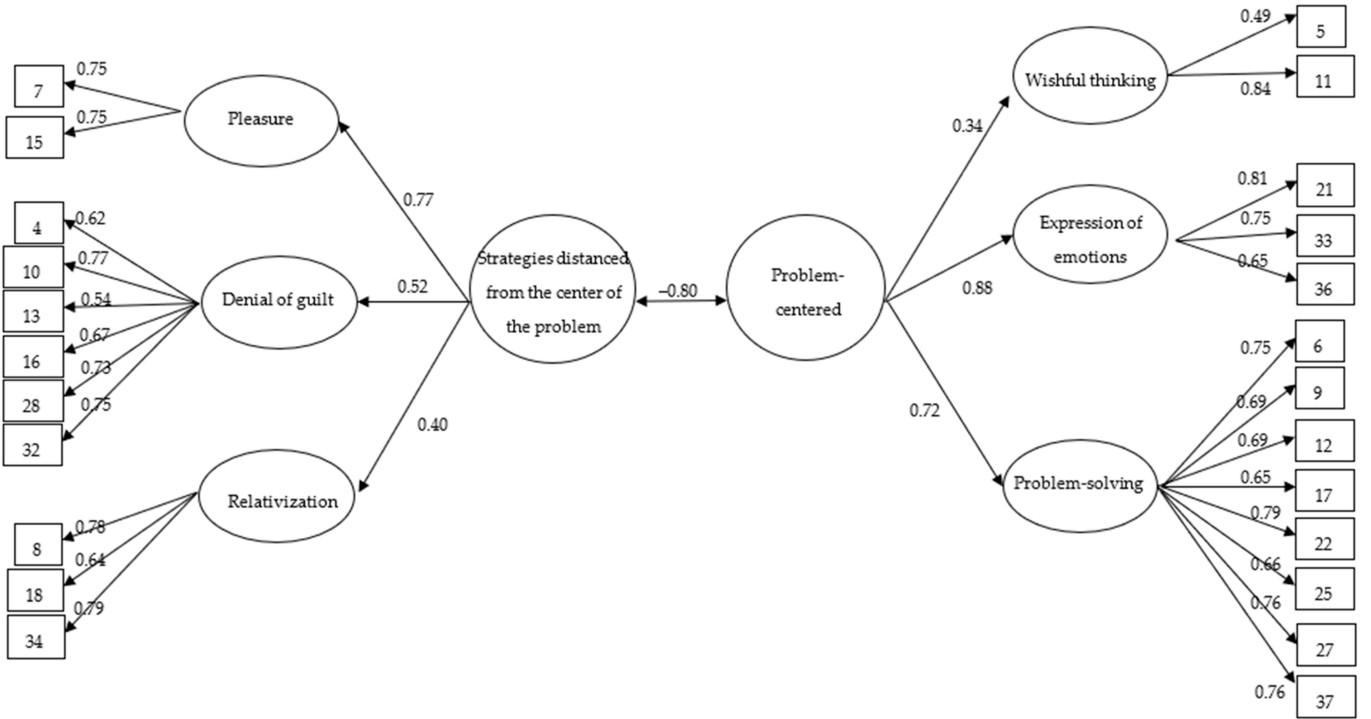

**Figure 1.** Testing of the two-dimensional model of coping $\chi^2$ (231) = 475.61, $p < 0.01$; $\chi^2/df$ = 2.06; GFI = 0.90.; CFI = 0.93; RMSEA = 0.05; SRMR = 0.07; The circles indicate latent variables, and the boxes indicate the number of the item. All paths are significant (1% level); N = 345.

### 3.2. Generalized Linear Models

Covariance analyses (ANCOVA) were carried out to study the influence of the experimental conditions on psychological distance in relation to CC (Table 2). The variables regarding age and gender were controlled. The control group was used as a reference for the estimates of parameter in order to achieve the covariance analyses and to analyze the influence of each experimental condition.

**Table 2.** Means and analysis of covariances results of experimental groups.

| Dependant Variables | Proximal Conditions (N = 88) | | | | | | | | Distal Conditions (N = 87) | | | | | | | | Control Group (N = 108) | | $F_{(8, 278)}$ | $p$ | $\eta p^2$ |
| | Social | | Spatial | | Temporal | | Uncertainty | | Social | | Spatial | | Temporal | | Uncertainty | | | | | | |
| | M | SD | M | SD | M | SD | M | SD | M | SD | M | SD | M | SD | M | SD | M | SD | | | |
|---|---|---|---|---|---|---|---|---|---|---|---|---|---|---|---|---|---|---|---|---|---|
| Psychological distance | 1.70 | 0.43 | 1.88 | 0.46 | 1.66 | 0.35 | 1.73 | 0.46 | 1.93 | 0.37 | 1.90 | 0.43 | 1.86 | 0.47 | 2.09 | 0.43 | 1.81 | 0.39 | 2.45 | 0.01 ** | 0.07 |
| Social and temporal barriers | 1.47 | 0.33 | 1.54 | 0.53 | 1.38 | 0.37 | 1.33 | 0.37 | 1.33 | 0.29 | 1.45 | 0.53 | 1.45 | 0.45 | 1.62 | 0.51 | 1.39 | 0.37 | 1.67 | 0.10 | 0.05 |
| Spatial barrier | 1.73 | 0.65 | 2.13 | 0.72 | 1.94 | 0.83 | 2.35 | 1.19 | 2.70 | 0.97 | 2.38 | 0.97 | 2.17 | 0.98 | 2.35 | 2.27 | 2.16 | 0.86 | 1.95 | 0.05 * | 0.05 |
| Uncertainty | 2.13 | 0.73 | 2.31 | 0.79 | 1.97 | 0.57 | 1.93 | 0.68 | 2.36 | 0.78 | 2.33 | 0.66 | 2.34 | 0.66 | 2.86 | 0.59 | 2.32 | 0.67 | 3.19 | 0.00 ** | 0.09 |
| Problem-centered strategies | 3.15 | 0.65 | 3.20 | 0.72 | 3.10 | 0.67 | 3.38 | 0.76 | 3.58 | 0.45 | 3.54 | 0.64 | 3.15 | 0.72 | 3.19 | 0.76 | 3.26 | 0.72 | 1.19 | 0.31 | 0.03 |
| Strategies distanced from the center of the problem | 2.77 | 0.50 | 2.66 | 0.38 | 2.72 | 0.46 | 2.62 | 0.42 | 2.59 | 0.36 | 2.68 | 0.45 | 2.63 | 0.39 | 2.72 | 0.41 | 2.64 | 0.50 | 0.26 | 0.98 | 0.00 |
| Problem-solving | 3.37 | 0.79 | 3.50 | 0.84 | 3.37 | 0.81 | 3.69 | 0.93 | 3.74 | 0.65 | 3.71 | 0.76 | 3.42 | 0.92 | 3.47 | 0.87 | 3.50 | 0.84 | 0.37 | 0.93 | 0.01 |
| Expression of emotions | 2.33 | 0.89 | 2.55 | 1.00 | 2.32 | 1.01 | 2.50 | 0.87 | 3.02 | 0.64 | 2.86 | 0.88 | 2.36 | 0.97 | 2.38 | 1.08 | 2.49 | 0.99 | 1.81 | 0.08 | 0.05 |
| Wishful thinking | 3.48 | 0.91 | 2.95 | 0.79 | 3.18 | 1.21 | 3.47 | 0.99 | 3.79 | 0.84 | 3.88 | 0.93 | 3.24 | 1.05 | 3.26 | 0.96 | 3.48 | 1.02 | 2.16 | 0.03 | 0.06 |
| Relativization | 2.35 | 0.90 | 2.40 | 0.75 | 2.54 | 0.85 | 2.40 | 0.77 | 2.56 | 0.86 | 2.81 | 0.75 | 2.43 | 0.82 | 2.68 | 0.73 | 2.60 | 0.91 | 0.73 | 0.66 | 0.02 |
| Denial of guilt | 2.59 | 0.68 | 2.58 | 0.66 | 2.52 | 0.79 | 2.34 | 0.71 | 2.44 | 0.48 | 2.40 | 0.84 | 2.46 | 0.57 | 2.48 | 0.72 | 2.39 | 0.83 | 0.36 | 0.94 | 0.01 |
| Pleasure | 3.49 | 0.81 | 3.34 | 0.68 | 3.49 | 0.90 | 3.41 | 0.79 | 3.11 | 0.64 | 3.45 | 0.59 | 3.19 | 0.70 | 3.32 | 0.64 | 3.40 | 0.77 | 0.50 | 0.86 | 0.01 |
| Resgnation | 2.99 | 0.76 | 2.64 | 0.53 | 2.77 | 0.68 | 2.79 | 0.72 | 2.59 | 0.62 | 2.62 | 0.66 | 2.77 | 0.68 | 2.80 | 0.54 | 2.67 | 0.83 | 0.75 | 0.65 | 0.02 |

** $p < 0.01$; * $p < 0.05$.

Generally speaking, belonging to experimental groups enables the prediction of the psychological distance scores (H2a). More precisely, belonging to experimental groups allows prediction of the scores obtained on the scale of psychological distance from CC ($F(8278) = 2.45$, $p < 0.01$, $\eta_p^2 = 0.07$), and more specifically the scores obtained on the spatial aspect dimension ($F(8, 278) = 1.95$, $p = 0.05$, $\eta_p^2 = 0.05$), and on the dimension related to the uncertain nature of the object ($F(8, 278) = 3.19$, $p < 0.01$, $\eta_p^2 = 0.09$). More precisely, the conditions relative to the temporal aspect and the uncertainty are those that enable the scores relative to the uncertain nature of CC to be explained (respectively: B = −1.16, *SE* = 0.50, *p* < 0.05, B = −1.31, *SE* = 0.60, *p* < 0.05). Lastly, the condition linked to the social aspect enables the scores obtained on the spatial dimension to be explained (B = −1.33, *SE* = 0.64, *p* < 0.05).

In a second phase, we examine all the conditions presenting greater psychological distance in relation to CC. Together, the four conditions (temporal, social, spatial, and uncertainty aspects) permit prediction of the scores on the psychological distance scale (B = 1.63, *SE* = 0.76, *p* < 0.05). More specifically, the fact of watching a video presenting victims of CC (social aspect condition) explains the scores obtained on the spatial barrier (B = 1.58, *SE* = 0.68, *p* < 0.05). Lastly, the condition of the uncertain nature of CC enables explanation of the scores obtained with regard to the uncertain nature of CC (B = 1.67, *SE* = 0.57, *p* < 0.01).

The experimental conditions enable the scores of only one coping strategy to be explained (H2b). Thus, belonging to the experimental groups allows prediction of the scores relative to wishful thinking only ($F(8, 278) = 2.16$, $p = 0.03$, $\eta_p^2 = 0.06$). More precisely, the fact of watching a video presenting the nearby physical impacts of CC (spatial aspect condition) explains the scores obtained on wishful thinking (B = −1.13, *SE* = 0.52, *p* < 0.05).

*3.3. Path Analysis*

In order to identify the relations between psychological distance and coping strategies, while taking into account experimental conditions, two causal research models were tested (H2c). In order to test those models, two modalities were considered, one in which the individuals watched a video presenting CC as being close (whatever the aspect emphasized)(*n* = 88, encoded 0), and one in which they watched a video presenting CC as distant (whatever aspect is emphasized, encoded 1)(*n* = 87). In order to verify the model's adjustment, the recommendations presented above were followed [50–52]: $\chi^2 = ns$ or $\chi^2/df$ between 1 and 3; GFI > 0.90; CFI > 0.90; RMSEA < 0.08; SRMR < 0.08.

For the first model, the two coping sets identified above were considered: Problem-centered coping and coping distanced from the center of the problem (Figure 2). This model is correctly adjusted: ($\chi^2 (8) = 17.21$, $p = 0.03$; $\chi^2/df = 2.15$; GFI = 0.97; CFI = 0.94; RMSEA = 0.08; SRMR = 0.06). Figure 2 shows the links between the type of experimental condition and the scores of the dimensions of psychological distance relative to the spatial barrier (β = 0.18, *p* < 0.01) and of the uncertain nature (β = 0.28, *p* < 0.01). Furthermore, the scores regarding the spatial dimension and the hypothetical nature of the object allow those relative to the social and temporal barriers to be explained (respectively: β = 0.18, *p* < 0.01 and β = 0.37, *p* < 0.01). Lastly, only the scores of the dimension in relation to the social and temporal barriers can give rise to an explanation of those of the problem-centered coping (β = −0.58, *p* < 0.01) and of coping strategies that are at a distance from the center of the problem (β = 0.25, *p* < 0.01).

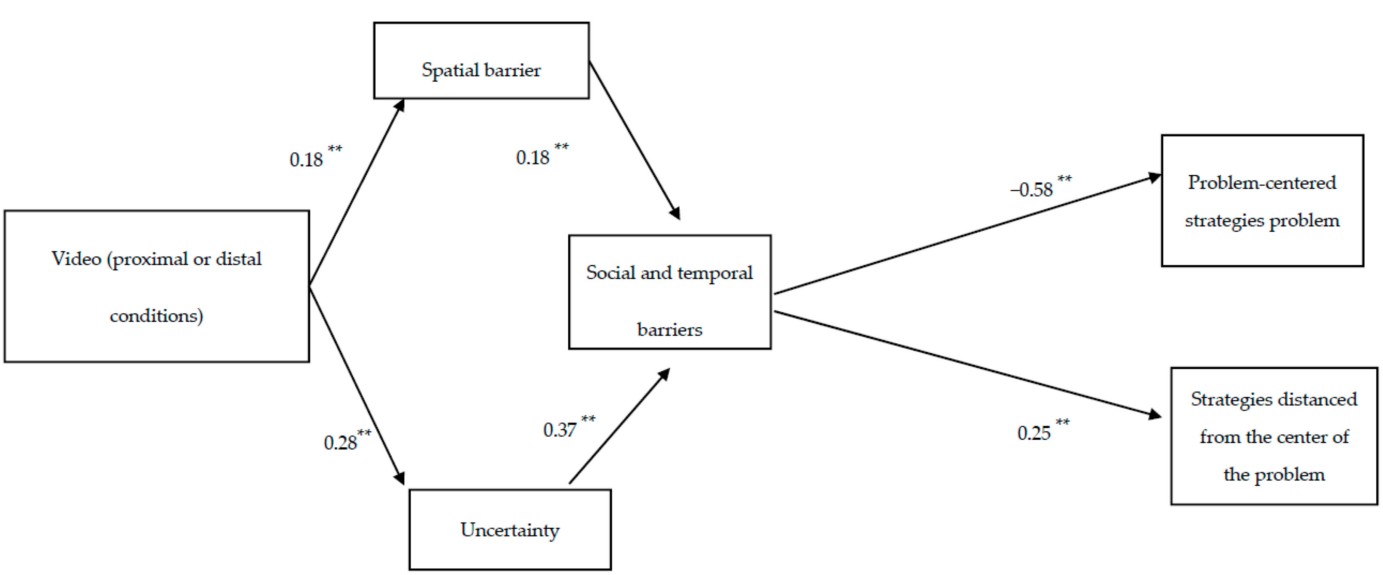

**Figure 2.** Path analysis model taking into account the two global coping strategies. $\chi^2$ (8) = 17.21, $p$ = 0.03; $\chi2/df$ = 2.15; CFI = 0.94; GFI = 0.97; RMSEA = 0.08; SRMR = 0.06; Video: Proximal conditions coded 0, and distal conditions coded 1; ** $p < 0.01$; * $p < 0.05$.

The second model tested corresponds to the same model presented above, with the exception that we replaced the two general coping sets (e.g., problem-centered strategies) by the various coping strategies (e.g., expression of emotions, problem-solving, wishful thinking). This second model also illustrates a causal path, by linking the experimental conditions with the scores relative to the psychological distance dimensions, the latter being in relation with the coping strategies (Figure 3). A good adjustment of the model is observed: $\chi^2$ (27) = 36.34, $p$ = 0.11; $\chi^2/df$ = 1.35; GFI = 0.96; CFI = 0.96; RMSEA = 0.05; SRMR = 0.07. This model highlights significant relations between the type of video watched and the scores on the spatial barrier ($\beta$ = 0.18, $p. < 0.01$) and on the uncertainty barrier ($\beta$ = 0.28, $p < 0.01$). The latter two dimensions enable an explanation of the scores obtained on the dimension relative to the social and temporal barriers (respectively: $\beta$ = 0.18, $p < 0.01$ and $\beta$ = 0.37, $p = 0.01$). Then, the scores obtained on the uncertain nature of the object allow prediction of the relativization scores ($\beta$ = 0.18, $p < 0.01$) and those linked to problem-solving ($\beta$ = −0.14, $p < 0.05$). Lastly, the scores on the dimension relative to the social and temporal barriers enable explanation of those of the denial of guilt ($\beta$ = 0.22, $p < 0.01$), of pleasure ($\beta$ = 0.17, $p < 0.05$), of relativization ($\beta$ = 0.18, $p < 0.01$), of wishful thinking ($\beta$ = −0.21, $p < 0.01$), of expression of emotions ($\beta$ = −0.43, $p < 0.01$), and of problem-solving ($\beta$ = −0.55, $p < 0.01$).

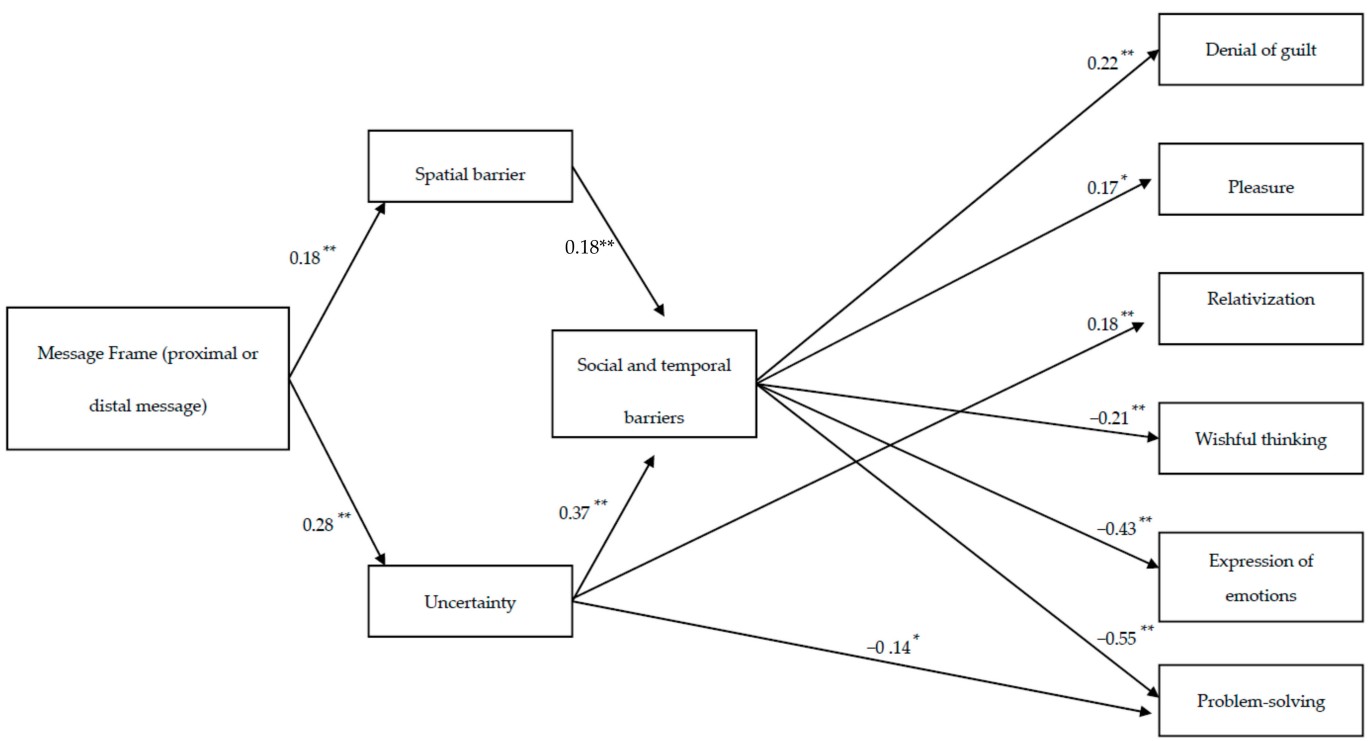

**Figure 3.** Path analysis model taking into account the six coping strategies. $\chi^2$ (27) = 36.34, $p$ = 0.11; $\chi^2/df$ = 1.35; CFI = 0.96; GFI = 0.96; RMSEA = 0.05; SRMR = 0.07; Video: Proximal conditions coded 0, and distal conditions coded 1; ** $p$ < 0.01; * $p$ < 0.05.

## 4. Discussion

As a reminder, this study was set up in order to reach two objectives. The first aims to identify the coping strategies implemented in the face of the phenomenon of CC. The second objective is to analyze the influence of psychological distance relative to CC on coping strategies.

### 4.1. Two Sets of Coping Strategies Linked to the Issue of Climate Change

The hypothesis whereby coping in relation to CC is composed of two large sets of strategies is confirmed (H1). Indeed, as in the work of Homburg et al. [35] on global environmental problems (species extinction, desertification, CC, etc.), we identify two second-order factors: Strategies centered on the problem and others at a distance from the core of the problem. As suggested in the literature, our results clearly bring to light two sets of strategies: One centered on active coping (problem-centered) and one centered on passive coping (at a distance from the problem) [35,39,41,42]. More specifically, our analyses show that wishful thinking explains the same second-order factor as problem-solving and expression of emotions. It thus appears that it is also a problem-centered strategy in the context of the issue of climate change. Indeed, this wishful thinking illustrates a strategy where the individual does not minimize the gravity of CC and expresses the fact that it is an issue that needs to be handled.

On the one hand, problem-centered coping seems to involve strategies whereby the individual accepts the issue of CC. Thus, they admit that CC exists by trying to resolve the situation (problem-solving), by thinking about it (expression of emotions), and/or by hoping that it resolves itself (wishful thinking). The scientific literature highlights the fact that active coping refers to a situation that seems controllable [34,37]. On the other hand, coping strategies that are distanced from the problem are avoidance strategies where the individual minimizes the gravity of CC and opts out. The strategies related to this type of coping apparently illustrate that the individual does not feel responsible for the situation (denial of guilt); they minimize the gravity of the phenomenon (relativization);

and it does not stop them from living (pleasure). These types of strategies would notably be implemented when the individual does not believe they can remedy this situation [41,42]. These elements suggest that the threatening and/or controllable nature of CC could lead to the adoption of certain coping strategies.

### 4.2. Influence of the Experimental Conditions on Psychological Distance and Coping Strategies

#### 4.2.1. Influence of Experimental Conditions on Psychological Distance

Our hypothesis whereby experimental conditions influence psychological distance relative to CC is confirmed (H2a). Our results corroborate with those of the studies showing the influence of a message on the manner of assessing CC [23,28].

On the one hand, the fact of presenting CC as concrete seems to lead individuals to question less the relation between the phenomenon and its effects. When CC is presented as being certain and/or currently underway, individuals perceive its effects more, at a low spatial level. Because the fact of not questioning CC leads to more concerns [14], it is possible that individuals feel more concerned by the effects at a local level and perceive them more. On the other hand, presenting CC as distant is also associated to an abstract representation of the phenomenon. Featuring the victims of CC living in precariousness (social aspect, large distance) leads individuals to perceive the effects of CC as being more distant from a spatial point of view. Thus, it is possible that large social distance implies large spatial distance. In other words, individuals could consider that CC impacts people who do not belong to their social groups, and impacts regions other than their own [54]

These various elements emphasize the interdependence between the psychological distance barriers. In particular, we observe the grouping of the social and temporal barriers, which can be explained by the fact that they refer to merged elements. In other words, projecting oneself into the future can also relate back to different social groups. In this sense, research has shown that individuals can process information in the same way when the social and/or temporal aspect of the object in question is manipulated [7,55,56].

#### 4.2.2. Influence of the Experimental Conditions on Coping Strategies

We had supposed that the experimental conditions would influence coping strategies relative to CC (H2b), but this hypothesis is only partially confirmed.

When we consider the experimental conditions independently from one another, no significant direct influence on the scores of coping strategies is observed, apart from the one relative to wishful thinking. Indeed, the condition in which we present the effects of CC at a low spatial level is associated with lower scores of wishful thinking, considered as a problem-centered strategy. Thus, when we present the effects of CC in mainland France, individuals declare less wishful thinking regarding the resolution of the situation. Presenting the phenomenon as being spatially close leads individuals to being less optimistic regarding the situation. As the object is more concrete, the threat and/or gravity of the situation are potentially more clearly perceived [20,21]. In this sense, the situation could seem less controllable when CC is presented as being spatially close. The role of the spatial barrier is unclear and more research is needed [22]. Furthermore, alarmist messages would not be associated with better individual adaptation [57]. Indeed, alarmist messages and stress would reduce attentional deployment [58]. Simultaneously, analysis of the research causal models highlights links between the experimental conditions, referring back to the fact of having seen a video presenting CC as being close or distant (whatever the psychological distance aspect), psychological scores, and coping scores. More precisely, it seems that the psychological distance barriers mediate the relation between experimental conditions and coping strategies.

Research studies show that the use of coping strategies depends on the controllable nature of the situation and on the fear that it generates [34,39,41,47]. In order to better understand the influence of psychological distance barriers on coping strategies, it would be pertinent to consider complementary elements such as responsibility, fear, and perceived controllability.

### 4.2.3. Relations between Experimental Conditions, Psychological Distance Barriers and Coping Strategies

Analyses of the causal research models show that the manner in which CC is presented influences psychological distance regarding CC, which itself impacts coping strategies. Significant relations between the psychological distance barriers and coping strategies are observed, thus confirming our hypothesis (H2c). It is noted that the experimental conditions directly influence the spatial barrier and the uncertain nature of CC. While the scores regarding the uncertain nature of the phenomenon enable rationalization and problem-solving to be explained, the social and temporal barriers scores explain all the coping strategies. Thus, the closer CC is perceived as being from a temporal and social point of view, the more problem-centered strategies are declared, and the less strategies distanced from the problem are declared. These results corroborate with those of other research showing that perceiving CC as concrete leads individuals to adapt to the environmental situation [23,24].

These elements highlight the pertinence of considering the influence of each of the psychological distance barriers, beyond the consideration of psychological distance in a uni-dimensional manner. These results also go in the direction of improved inclusion of specific coping strategies. Indeed, the analyses performed show that a small spatial distance can be associated to less wishful thinking, one of the problem-centered coping strategies. It is thus possible that this condition, where the spatial aspect of CC is emphasized, influences problem-centered coping strategies. It seems necessary to consider further measures during future research, in order to improve understanding of these relations. For example, Trope and Liberman [7] underscore the necessity of considering the controllability of a situation according to the various aspects of psychological distance. Once again, it seems pertinent to study perceived threat in that this element is linked to coping strategies and the feeling of stress [42].

When the various types of coping strategies are taken into account, it is observed that the scores on the social and temporal barriers enable an explanation of denial of guilt, pleasure, relativization, wishful thinking, expression of emotions, and problem-solving. These relations are negative with problem-centered strategies and positive with those distanced from the problem. Thus, the fact of perceiving CC as being close on social and temporal levels would lead the individual to accept the phenomenon more and minimize it less. Likewise, perceiving CC as an incontestable phenomenon could lead individuals to use strategies linked to problem-solving, and to declare fewer relativization strategies. Thus, when individuals do not challenge the phenomenon, they adapt more to it [26,59].

### 4.3. Limits

Even if this study confirms relations between psychological distance and coping strategies, its limits need to be pointed out. First of all, several situations are presented according to the participant's condition. According to the focused psychological distance barrier, videos thus show different situations that may vary in terms of perceived gravity or emotional arousal. In future studies, other measures and manipulations checks should be added to control these elements. Moreover, the sample sizes per experimental condition are too small to set up multi-group structural equation models. In order to consider the experimental conditions in our models, we had to create a binary variable in relation to these experimental conditions. Thus, the latter were not treated independently from one another, as they were grouped according to the fact that CC was presented as close or distant, whatever the aspect emphasized. It would be interesting to recruit more participants during future research in order to achieve better understanding of the influence of each of these psychological distance barriers. Furthermore, the sample is not representative, and we did not control if the participants were aware of the CC issue or if there are self-selection bias. Indeed, it would have been pertinent to include how aware the individuals were of this object in the analysis of the results. Lastly, the factorial analysis did not allow us to identify the temporal and social aspects of CC independently. It would be worthwhile to

conduct interviews among individuals in order to identify the particularities of each of these two dimensions better, in order to improve this tool.

## 5. Conclusions

This study enabled two large sets of coping strategies in relation to CC to be identified. One of them refers to problem-centered coping and seems to be linked to acceptance of the environmental issue, and the other concerns coping strategies that are at a distance from the center of the problem, and rather seems to illustrate the minimization of the issue's gravity. The results highlight the relations between psychological distance in relation to CC and the various environmental coping strategies. These results also show the interdependence between the psychological distance barriers and the way they predict coping. In order to fill the gaps of this research, it would be interesting to study to what extent individuals perceive CC and its effects as a threat, and how they adapt to it. In order to do so, the perception of the risks relative to CC could be studied according to the angle of the psychometric paradigm [60]. Lastly, this study enables us to advance the idea that it seems pertinent to consider the various aspects of psychological distance in order to understand the individual reactions when facing environmental awareness campaigns (acceptance or rejection). In order to encourage individuals to take an interest and adapt to the phenomenon, it seems pertinent to communicate about the concrete aspects of climate change by relying on these psychological distance barriers.

**Supplementary Materials:** The supplementary is available online at https://www.mdpi.com/2071-1050/13/2/992/s1.

**Author Contributions:** Conceptualization, M.G., G.F.-B., and O.N.; methodology, M.G., G.F.-B., and O.N.; software, SPSS 23.; validation, M.G., G.F.-B., and O.N.; formal analysis, M.G., G.F.-B., and O.N.; investigation, M.G.; resources, N/A.; data curation, M.G.; writing—original draft preparation, M.G., G.F.-B., and O.N.; writing—review and editing, M.G.; visualization, N/A.; supervision, G.F.-B. and O.N.; project administration, N/A.; funding acquisition, N/A. All authors have read and agreed to the published version of the manuscript.

**Funding:** This research received no external funding.

**Institutional Review Board Statement:** Not applicable given this is a non-interventional research.

**Informed Consent Statement:** Informed consent was obtained from all subjects involved in the study.

**Data Availability Statement:** Data presented in this manuscript are available upon request from the corresponding author.

**Conflicts of Interest:** The authors declare no conflict of interest.

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
