# Peer review of "Encouraging Individuals to Adapt to Climate Change: Relations between Coping Strategies and Psychological Distance"

_sustainability, doi:10.3390/su13020992_

Round 1

Reviewer 1 Report

This paper constitutes an interesting and quite ambitious article which successfully checks two broad kinds of hypotheses. The theoretical framework is confusing at some moments and should be expanded at some points. Language revision should be addressed.

30-31. “Given its anthropogenic cause, it seems essential for individuals to mobilize in order to adapt to this environmental phenomenon”. Individuals should adapt because of the seriousness of the problem, not because it is of anthropogenic origin. Unless authors mean that individuals should change (not adapt) their behaviours because those are the origin of climate change.

32-33. “In this regard, 93% of French people say that global warming is in progress and is caused by human activities”. I don’t know that survey, but I hope it is representative. I guess they asked about global warming instead of climate change but global warming and climate change are not synonymous. It would be more useful using another (representative) survey asking specifically about climate change in France.

  1. “Regarding environmental situations”. Confusing. I guess authors mean “environmental problems”. If not, an example is required.
  2. “In addition, many individuals do not seem to question the uncertain nature of CC”. This line seems rather disconnected from the rest of the paragraph. A couple of extra lines developing this idea would be helpful. Probably expand this idea and making it the beginning of the following paragraph.
  3. Again, talking of “environmental phenomena” is confusing. Better talk of environmental risks or hazards, or problems instead of phenomena.
  4. “Psychological distance barriers”. First time authors introduce this concept. It should have been defined or explained before.
  5. “at a low spatial level”. Confusing.
  6. “sentiment of environmental preoccupation” environmental concern
  7. “The social aspect of CC”. It is unclear what authors want to mean.
  8. “pro-environmental engagement”. Which kind of engagement? Authors mean attitudes or behaviours?
  9. “when CC is perceived as real”. Perceive climate change as a distant problem does not mean individuals think climate change is not real. Put it like this, this sentence is an oxymoron
  10. “some authors suggest that perceiving proximal CC could be inefficient”. Inefficient in which sense?

On the whole the first paragraph of the 1.2. section is highly confusing mixing different concepts (many of them are undefined) which in some cases don’t seem much related to the aim of the paper.  I am dubious that this first paragraph can cast a conclusion such as “This lack of consensus highlights the relevance to study in what extent concreteness of a situation”. Authors should focus their efforts to disentangle theoretical background to finally cast that conclusion that justifies their research. Also the four dimensions of psychological distance should be clearer defined.

  1. “if this adaptation is based on the problem or not”. I’m probably wrong, but I find difficult to picture an adaptation which is not based in a problem. Probably an example or an extension of this idea could be of use. Or maybe if it is not very important it could be discarded.
  2. “This research study”. Authors may choose any of the two words instead
  3. “in order to identify the influence of each of the dimensions of psychological distance”. Authors should state clearer which one their dependent variable is.
  4. The paragraph should end with a shorter and clearer hypothesis. I think “also” in 133 line is problematic. In all cases, a final sentence (probably in italics) summarising and presenting each hypothesis, short and clear, would be of much use for explanatory purposes.
  5. Authors probably don’t need to explain at that stage how are they going to check their hypotheses.
  6. It would be useful to know if besides place of residence, age was considered as an exclusion item in order to make up the sample. Under 18 years old were included? Also the dates of the survey should be stated.
  7. 80% of female respondents seems a quite high percentage. I wonder if it is usual in this kind of survey and, if not, if authors can explain it. It makes me think that maybe the way of publishing or the sites or the “snowball effect” somehow created or favoured a self-selection bias. Also, the percentage should include decimals as the following figure in order to make the 100%.
  8. The examples provided are very useful. Also the availability of the experimental videos on Youtube is a good idea.
  9. I think is unnecessary
  10. Text becomes easier to follow if authors explicitly cite the four items. I’m not quite sure that “withdrawal” is the correct expression there.

Fig. 1. I guess the numbers in boxes correspond to the items in the questionnaire. It should be stated clearly in the figure. And if possible, those items should be included in a annexe.

317-318. Confusing sentence.

324-326. Confusing sentence.

327-328. Confusing sentence.

Table 2. Asterisks are useless in the legend. But the level of signification used (0.05 probably) should be stated. It must be only one level of signification because we don’t have two kinds of bold to make a differentiation between 0.01 and 0.05.

Fig. 2. Here the asterisks legend is needed.

  1. Confusing paragraph.

Author Response

Response to Reviewer 1 Comments

Point 1: 30-31. “Given its anthropogenic cause, it seems essential for individuals to mobilize in order to adapt to this environmental phenomenon”. Individuals should adapt because of the seriousness of the problem, not because it is of anthropogenic origin. Unless authors mean that individuals should change (not adapt) their behaviours because those are the origin of climate change.

 Response 1: modification L 30-31

Point 2: 32.33. “In this regard, 93% of French people say that global warming is in progress and is caused by human activities”. I don’t know that survey, but I hope it is representative. I guess they asked about global warming instead of climate change but global warming and climate change are not synonymous. It would be more useful using another (representative) survey asking specifically about climate change in France.

Response 2: The cited survey is representative, but we propose modificationsL 32-33

Point 3: 46. “Regarding environmental situations”. Confusing. I guess authors mean “environmental problems”. If not, an example is required.

Response 3: modification L. 54

Point 4: 47. “In addition, many individuals do not seem to question the uncertain nature of CC”. This line seems rather disconnected from the rest of the paragraph. A couple of extra lines developing this idea would be helpful. Probably expand this idea and making it the beginning of the following paragraph.

 Response 4: lines 58 – 63 have been modified  

Point 5: 48. Again, talking of “environmental phenomena” is confusing. Better talk of environmental risks or hazards, or problems instead of phenomena.

Response 5: modification L. 64

Point 6: 49. “Psychological distance barriers”. First time authors introduce this concept. It should have been defined or explained before.

Response 6: modification L 53 is proposed 

Point 7: 50. “at a low spatial level”. Confusing.

Response 7: modification lines 81-84

Point 8: 51. “sentiment of environmental preoccupation” environmental concern

Response 8: modification line 83

Point 9: 52. “The social aspect of CC”. It is unclear what authors want to mean.

Response 9: modification lines 84 - 86

Point 10: 53. “pro-environmental engagement”. Which kind of engagement? Authors mean attitudes or behaviours?

Response 10: Replace with “…to more pro-environmental intentions” line 88.

Point 11: 54. “when CC is perceived as real”. Perceive climate change as a distant problem does not mean individuals think climate change is not real. Put it like this, this sentence is an oxymoron

Response 11: modification - lines 90 - 93

Point 12: 55. “some authors suggest that perceiving proximal CC could be inefficient”. Inefficient in which sense?

 Response 12: modification - lines 93 - 95

Point 13: On the whole the first paragraph of the 1.2. section is highly confusing mixing different concepts (many of them are undefined) which in some cases don’t seem much related to the aim of the paper.  I am dubious that this first paragraph can cast a conclusion such as “This lack of consensus highlights the relevance to study in what extent concreteness of a situation”. Authors should focus their efforts to disentangle theoretical background to finally cast that conclusion that justifies their research. Also the four dimensions of psychological distance should be clearer defined.

Response 13: Several sentences (see before) have been modified in order to be clearer + Modifications – lines 105 - 107

Point 14: 92. “if this adaptation is based on the problem or not”. I’m probably wrong, but I find difficult to picture an adaptation which is not based in a problem. Probably an example or an extension of this idea could be of use. Or maybe if it is not very important it could be discarded.

Response 14: Modify : “This lack of consensus highlights the relevance to study in what extent psychological distance or proximity are associated with individual adaptation to climate change, or not. In that sense, it seems interesting to consider adaptation to face an environmental threat.”

Point 15: 93. “This research study”. Authors may choose any of the two words instead

Response 15: Modification line 141

Point 16: 94. “in order to identify the influence of each of the dimensions of psychological distance”. Authors should state clearer which one their dependent variable is.

Response 16: modification ; lines 143 - 146

Point 17: 95. The paragraph should end with a shorter and clearer hypothesis. I think “also” in 133 line is problematic. In all cases, a final sentence (probably in italics) summarising and presenting each hypothesis, short and clear, would be of much use for explanatory purposes.

Response 17 : modifications, L146-149

Point 18: 96. Authors probably don’t need to explain at that stage how are they going to check their hypotheses.

 Response 18: this following sentence has been removed : « This last hypothesis will be tested with path analyses, in order to take into account the influence of the experimental conditions".  line 176

Point 19: 97. It would be useful to know if besides place of residence, age was considered as an exclusion item in order to make up the sample. Under 18 years old were included? Also the dates of the survey should be stated.

 Response 19: modifications : L 181 - 183

Point 20: 98. 80% of female respondents seems a quite high percentage. I wonder if it is usual in this kind of survey and, if not, if authors can explain it. It makes me think that maybe the way of publishing or the sites or the “snowball effect” somehow created or favoured a self-selection bias. Also, the percentage should include decimals as the following figure in order to make the 100%.

Response 20: percentage was modified (L184). Point 4.3 « Limits », the following sentence was modified in order to talk about selection bias : “Furthermore, the sample is not representative, and we did not control if the participants were aware of the CC issue or if there are self-selection bias” 

Point 21: 99. The examples provided are very useful. Also the availability of the experimental videos on Youtube is a good idea.

Response 21: ok, thanks to the reviewer.

Point 22: 100. I think is unnecessary

Response 22: unfortunately, i cannot find which information is unnecessary. I think the reviewer has made a mistake with lines numbers...

Point 23: 101. Text becomes easier to follow if authors explicitly cite the four items. I’m not quite sure that “withdrawal” is the correct expression there.

Response 23: modification : L266-269, and L281-287 ; items have been added in the appendix

Point 24: Fig. 1. I guess the numbers in boxes correspond to the items in the questionnaire. It should be stated clearly in the figure. And if possible, those items should be included in a annexe.

Response 24: modification L 353-354 ; items have been added in the appendix

Point 25: 317-318. Confusing sentence. (L 377 for us)

Response 25: modification L 377-379

Point 26: 324-326. Confusing sentence.(L384 for us)

Response 26: modification 384-386

Point 27: 327-328. Confusing sentence. (L387 for us)

Response 27: the sentence have been removed

Point 28: Table 2. Asterisks are useless in the legend. But the level of signification used (0.05 probably) should be stated. It must be only one level of signification because we don’t have two kinds of bold to make a differentiation between 0.01 and 0.05.

Response 28: In order to be coherent with the legend, asterisks were added in the table.

Point 29: Fig. 2. Here the asterisks legend is needed.

Response 29: the legend was completed, table 2, pp. 11-12

Point 30: 437. Confusing paragraph.

Response 30: I think it is clearer now, with our modifications L81-86, point 1.2 and the items in appendices pp. 25-27

Reviewer 2 Report

The topic of the article is very actual for nowadays society. Below you will find few comments according content of the article:

1) The problem which is analysed in the article should be described more clearly. Now it is the impression that article is about two different phenomenon, two different problems.

2) the concepts which are analysed should be described. For example, every coping strategy should be described in literature analysis.

3) could authors of the article describe what the meaning is to show correlations of different coping strategies? How these results are helpful to reveal the aim of the research?

Author Response

Response to Reviewer 2 Comments

Point 1: The problem which is analysed in the article should be described more clearly. Now it is the impression that article is about two different phenomena, two different problems.

Response 1: Several sentences have been modified in order to specify our purpose regarding climate change (30 - 33; 82 – 86; 142-147; point 3.2).

Point 2: the concepts which are analysed should be described. For example, every coping strategy should be described in literature analysis.

Response 2: Sentences have been modified in order to explain clearer the psychological distance barriers (53; 58-64; 81-86; 90-94). Concerning the coping strategies (pp. 25-27), the items of the scales have been added in the appendix. A sentence has been added L 127-128.

Point 3: could authors of the article describe what the meaning is to show correlations of different coping strategies? How these results are helpful to reveal the aim of the research?

Response 3: Sentences have been modified in order to understand better the aims and the hypotheses of the study (145-148; point 1.4). The correlation scores show two second-order factors representing two families of coping strategies: problem-centered strategies, and strategies that are distanced from the center of the problem. These results lead us to answer to our hypothesis concerning coping strategies related to climate change.